# Development of New Biodegradable Agar-Alginate Membranes for Food Packaging

**DOI:** 10.3390/membranes12060576

**Published:** 2022-05-31

**Authors:** Sonia Amariei, Florin Ursachi, Ancuţa Petraru

**Affiliations:** Faculty of Food Engineering, Stefan cel Mare University of Suceava, 720229 Suceava, Romania; sonia@usm.ro (S.A.); ancuta.petraru@fia.usv.ro (A.P.)

**Keywords:** biodegradable packaging, biopolymers, ascorbic acid, calcium chloride

## Abstract

The paper analyzes the possibility of replacing the polyethylene packaging from food products with biodegradable packaging obtained from biopolymers. The proposed packaging materials were obtained from polysaccharides (alginate, agar), glycerol as plasticizer. To improve the properties necessary for the coating materials, two groups of membranes were made, one with ascorbic acid (AA, 0.1–0.45 g) in 150 mL filmogenic solution and the other with calcium chloride (CaCl_2_, 0.02–0.1 g) in 150 mL filmogenic solution. The membranes were analyzed for mechanical properties, light transmission, transparency and barrier properties (water vapor, oxygen, or fatty substances). The results demonstrated that the addition of AA (0.1 g), increases tensile strength, transparency, oxygen and water barrier properties. On the other hand, the addition of calcium chloride (0.08 g) increased the hardness, tensile strength and opacity of the membranes. Moreover, it ensured a uniform distribution of the mixture components. The uniformization of the mixture components in the presence of AA and CACl_2_ was observed by SEM and roughness analysis. Hydrogen bonding interactions between the biopolymers and the additives used were highlighted by FTIR analysis. All membranes have shown very good UV absorption. The results suggest that agar/alginate/glycerol membranes with AA and CaCl_2_ have the potential to be used in an active food packaging system.

## 1. Introduction

Food packaging is a major generator of plastic waste derived from conventional synthetic petroleum, which is neither biodegradable nor edible. Plastic is a basic material of the modern economy, being cheap, versatile, light and durable, transparent or opaque, and it is used in the most important sectors of any economy. The demand for plastic has surpassed all other basic materials such as steel, aluminum or cement, almost doubling since 2000, due to its excellent properties, exceeding at present more than 359 million tons [1]. Of this, nearly 40% is intended to be used as packaging [2,3]. The most important use of plastic in the EU is represented by packaging for dairy products, meat products, water bottles and soft drinks, and protective packaging for fruit [1]. The accumulation of these packages poses a threat to the environment, given that the equivalent of a truck loaded with plastic enters the ocean every minute, reduction in marine waste being a key factor in achieving the goal of sustainable development [3,4]. Currently, the increase in the number of elderly people and the decrease in the number of members in a family, especially in Europe, has led to a decrease in the amount of food purchased per day/week and an increase in the preference for small quantities, sliced products, packaged in trays and covered with polyethylene. This has increased the amount of non-degradable material with traces of food and created a real problem for the environment. Researchers are concerned with removing materials that are not biodegradable and replacing them by materials that are produced from proteins, polysaccharides, lipids or their blends, biomaterials, which are nontoxic and environmentally friendly [5], and that will form the matrix of films or membranes for food components.

The use of edible films and coatings for food packaging without active substances is also an active food packaging application, as the edibleness and biodegradability of films are additional functions that are not provided by conventional packaging materials [6].

In obtaining the membranes analyzed in this paper, sodium alginate and agar were used as polysaccharides and glycerol as plasticizer. Sodium alginate is a polysaccharide extracted from brown algae and its chain consists of β-D-mannuronic (M) acids and α-L-guluronic (G) acid sequences joined by glycosidic (1–4) bonds [7].

Agar obtained from the cell walls of some species of red algae is a mixture of two components, the linear polysaccharide agarose and smaller molecules of agaropectin [8,9,10]. Along with the biopolymers, the addition of plasticizer is necessary to reduce the fragility of films, to improve the flow and flexibility and to increase the hardness and to prevent them from cracking during packaging and transport [11].

Glycerol was chosen as a plasticizer, which is soluble in the chosen solvent (water) and it is miscible with the polymers in the composition [12]. The addition of glycerol plays an important role in disrupting the interactions between the molecular chains of polymers and rearranging them in another configuration, being characterized by the increase in the free volume between chains, enhancing flexibility, it also increases permeability to gases and vapors [13,14]. The barrier properties are extremely important especially for food packaging materials based on hydrophilic biopolymers. A low amount of glycerol that has a hydrophilic character will reduce the water vapor permeation [15]. A high transfer of water through membranes affects the quality of food and shortens their shelf life [16].

The use of calcium chloride (CaCl_2_) determines the formation of crosslink between Ca^2+^ ions and the two carboxyl groups on adjacent polymer chains in sodium alginate. In addition it improves the mechanical and the barrier properties of the membranes [17]. Calcium alginate has antiviral and antibacterial capacity [18], a very important property for food packaging materials. Immersion into a solution of CaCl_2_ or the addition of this substance to the composition of the polymer mixture is an important step in order to modify and control the barrier and mechanical properties of the membranes obtained [12]. Calcium chloride acts as a control system for residual or excessive moisture in packaging, thereby preventing or delaying the formation of microbial growth of films [19].

Ascorbic acid (AA) is a weak acid, soluble in water, with very good antioxidant properties and improves the color retention [20]. AA containing membranes that cover trays with meat preparations has the ability to prevent their lipid oxidation [16,21]. The moisturizing action of AA influences the recrystallization process of the mixture of biopolymers and plasticizer [22]. AA reduced the mechanical strength of the films, increased the transparency value of biopolymer films and improved the antimicrobial property of the membranes against *Escherichia coli (E. coli)* and *Staphylococcus aureus (S. aureus)* [20].

The aim of this work is to determine the most suitable concentrations of AA and CACl_2_ in the composition of the membranes that can be used in the packaging of food products. The main functions of this type of food packaging are to protect against the transfer of substances to and from food, against microorganisms, radiation and protection against mechanical action [3]. In the case of food products, UV radiation can cause the oxidation of lipid compounds, but there can also be an issue with the discoloration of meat products that makes them less attractive for consumers. The result of the research is a composition for an active food biodegradable membrane that is safe for the packaging of food products throughout the shelf-life period.

## 2. Materials and Methods

### 2.1. Films Development

The membranes were obtained by wet casting, when a hydrogel with a rearranged spatial structure was obtained, in which the biopolymers (agar, alginate) and the plasticizing agents used (glycerol and water) participate in various ratios. The membranes had a common matrix consisting of agar (1.25 g), alginate (3.00 g) and glycerine (0.75 g). AA (0.10/0.20/0.25/0.30/0.45 g) was added to one part of the samples (SAA1-5), and CaCl_2_ (0.01/0.02/0.04/0.08/0.1 g) to the others (SCa1-5). Each sample consisted of the mentioned components, to which ultrapure water up to 150 mL was added. AA and CaCl_2_ have been used as additives to improve the properties of membranes. Due to the intense reaction with Ca^2+,^ the cross-linking solution was added along with the other constituents to avoid local gelation. The mixture was homogenized by vigorous stirring (1400 rpm) using a magnetic stirrer (Heidolph MR Hei-Tec, Schwabach, Germany) at 90 °C for 30 min. The filmogenic solutions were cast onto a silicone foil and dried at room temperature (22–25 °C) for 2–3 days. Glycerol, sodium alginate, agar, AA and CaCl_2_ of analytical grade were provided by Carl Roth (Karlsruhe, Germany). After conditioning, the samples were kept in parchment paper envelopes, which were placed in a cardboard folder until the analyses were performed. All determinations were performed in the same laboratory, under the same conditions of temperature (24.5–25.2 °C) and relative humidity (31.0–32.0%).

### 2.2. Water Content (WC)

Film samples were weighed (W_1_), dried in an oven (Memmert, Schwabach, Germany) at 105 °C for 24 h, and weighted (W_2_) again. Water content (WC) was calculated with the relation [23]:(1)$$\mathrm{WC}=\frac{W_{1}-W_{2}}{W_{1}}\times 100$$

The results were expressed as grams of water per 100 g of sample.

### 2.3. Water Activity

The water activity values of the samples were determined using an AquaLab 4TE water activity meter (Meter Group, Pullman WA, USA). The determinations were made in triplicates [24].

### 2.4. Barrier Properties

The quality of food packaged in biopolymer membranes is influenced by membrane transfer processes between food and the atmospheric environment, such as water absorption, oxygen transfer, and loss of flavors or absorption of off-odors. To characterize the barrier properties of these packages, water vapor permeability, water vapor transmission rate, oxygen permeability, and oil resistance of the biodegradable films were determined [25].

#### 2.4.1. Water Vapor Permeability (WVP)

Acrylic cells with a diameter of 52 mm and a depth of 20 mm filled with calcium chloride (0% RH) up to a distance of 10 mm from the sealing membrane were placed in a desiccator containing saturated NaCl solution (75.0 ± 2% RH) [1,25,26]. Water vapor permeability (WVP) was determined gravimetrically with 0.0001 g accuracy in according to the ASTM E96-01 method. Determinations were made in triplicates at 0, 8, 24, 32 and 48 h and the WVP was calculated by Equation (2).
(2)$$\mathrm{WVP}=\left(\frac{w}{t}\right)\times \left(\frac{x}{\Delta P\times A}\right)$$
where, w/t (g/h) corresponds to the water mass absorbed by the system as against the time calculated by linear regression (R_2_ > 0.99) from weight data obtained during 48 h, A (m^2^) is the exposed film area, x (mm) is the thickness of the membrane, and ΔP (kPa) is the partial pressure difference through the film calculated by Equation (3).
(3)$$\Delta P=S\times \left(R_{1}\times R_{2}\right)\;$$
where, S is the saturated vapor pressure at 25 °C (3166 kPa), R_1_ and R_2_ are the [26] relative humidity in the desiccator (0.75) [27] and inside the cell (0.0), respectively, expressed in fractions [28].

#### 2.4.2. Water Vapor Transmission Rate (WVTR)

WVTR expressed in g/m^2^ h is calculated by the following Equation (4) [19,26]:(4)$$WVTR=\frac{w}{A\times t}\;$$
where, A is the area of the exposed film expressed in m^2^, w represents the amount of water in g transferred through the membrane and t is the time (expressed in h) of water transfer.

#### 2.4.3. Oxygen Barrier Property Measurement (OP)

The samples were cut into circles (Ø 3 cm) and sealed on top of an Erlenmeyer flask containing 3 g of sunflower oil. The flasks were held in a laboratory oven with air circulation ZRD-A5055 (Zhicheng, Analysis Instruments, Shangai, China) at 50 °C for five days [13].

Oxygen barrier property measurement was determined based on the peroxide value of the oil in accordance with the standard AOCS Cd 8-53 [29] with some modification. The determination of OP was based on the reaction of oxygen with the oil in Erlenmeyer flasks sealed with the membranes analyzed. The sunflower oil samples were treated with 20 mL mixture of glacial acetic acid and chloroform 2:1 and 1 mL saturated solution of potassium iodide. The control sample contained the same reagents, but oil free. The samples were kept in the dark for 3 min, after which 20 mL of distilled water was added and the iodine released was titrated by 0.01 sodium thiosulphate solution. The titration was continued until the light yellow color of the sample, after which 1 mL of starch solution was pipetted into each flask and titrated, until the blue color disappeared.

The peroxide index is calculated by the formula (Equation (5)):(5)$$\mathrm{PV}=\frac{\left(V-V_{0}\right)\times C\times 0.1269}{M}\times 100$$
where, V—is the volume of sodium thiosulphate used in the titration of the oil sample (mL), V_0_—the volume of sodium thiosulphate used in the titration of the control sample (mL), M—is the mass of oil used in determinations(g), C—is the concentration of sodium thiosulfate standard solution (mol/L), and 0.1269—is the mass of iodine (g) corresponding to 1 mL of sodium thiosulfate 0.01 m [13]. Since the amount of air enclosed in Erlenmeyer flasks will contribute to the oxidation of sample along with the oxygen transferred through membranes, an oil sample sealed in a flask with a polyethylene membrane, through which external oxygen does not penetrate, was used for comparison. The peroxide index of the closed PE sample was compared with that of the closed membrane samples, thus highlighting which of the membranes allows the transfer of oxygen from outside.

#### 2.4.4. Grease Permeability

The sunflower oil samples placed in 20 mL Erlenmeyer flasks were sealed with the membranes analyzed and placed upside down on the previously weighed filter paper. They were kept in the desiccator for 5 days, after which the filter paper was weighed to see if the membranes obtained allowed the passage of fatty substances.

#### 2.4.5. Film Thickness

The thickness of the membranes, expressed in micrometers, was determined by the Mitutoyo Corporation micrometer (Kawasaki, Tokyo, Japan), with a sensitivity of 0.001 mm. The final results were expressed as the average of ten thickness measurements [24].

### 2.5. Mechanical Properties

#### Tensile Strength (TS), Elongation at Break (E) and Hardness (H)

The texturometer Mark 10 Texture Analyzer loaded with a 5 kN cell (Mark 10 ESM301 Corporation, Copiague, NY, USA) was used for tensile strength (TS). Elongation at break (E) was determined according to ASTM standard, method D882-02. The conditioned films were cut into 10 mm × 125 mm pieces and mounted into the equipment with a stretching of 10 mm/min. The curves of force (N) as a function of deformation (mm) were recorded by the Texture Expert Exceed software. Tensile strength was calculated by dividing the force for film rupture by the area of the transverse section and elongation at break was calculated from the ratio of the increase in length to the original length, according to ASTMD 638-14 [13].

The Phase-II PHT-2500 Portable Hardness Tester was used to determine the hardness of the tested samples.

### 2.6. Optical Properties of the Films

#### 2.6.1. Color

The CIEL lab method was used to determine color, and the analyses were performed in triplicates with the Konica Minolta CM 700 Chromameter. The device was calibrated with a reference white porcelain tile (L_o_ = 97.63; a_o_ = 0.31 and b_o_ = 4.63), before the determinations. The illuminant used was D 65. The values of color space parameters L (lightness) ranging from zero (black) to 100 (white), a* ranging from +60 (red), to −60 (green) and b* (ranging from +60 (yellow), to –60 (blue) [30].

The color of films was expressed as the total color difference (Δ*E*) according to the following Equation (6):(6)$$\Delta E=\sqrt{{\left(\Delta L^{*}\right)}^{2}+{\left(\Delta a^{*}\right)}^{2}+{\left(\Delta b^{*}\right)}^{2}}$$
where, ΔE is the total color difference and ΔL∗, Δa∗, Δb∗—is the differential between the samples color parameter and the color parameter of the standard used as the film background [31,32].

#### 2.6.2. Film Opacity

Film opacity was measured according to the method described by Kalaycıoğlu et al., 2017 [33], using a Shimadzu 1800 UV spectrophotometer. The films were cut in rectangular pieces and were placed inside the test cell of the spectrophotometer. The blank cell was used as reference.

The UV-Vis spectra were recorded at wavelengths from 200 to 600 nm. The opacity was calculated by the following equation [34]:(7)$$\mathrm{Opacity}=\frac{A_{600}}{d}$$
where, A_600_ is the absorbance of the film at 600 nm and d is the film thickness in mm.

The transmittance values (%) were calculated at 600 nm. The transparency values were then calculated according to Han and Floros [13,35] as follows:(8)$$\mathrm{Transparency}=\frac{{\log\;T}_{600}}{x}$$
where, T_600_ is the transmittance (%) at 600 nm and x is the thickness of film samples (mm) [36].

### 2.7. Morphology

The surface and the cross-sectional morphologies of films were observed by scanning electron microscopy Tescan Vega II LMU (Tescan Orsay Holding, Brno, Czech Republic) The conditioned films were cut in small pieces, placed on a stub with an adhesive carbon tape, adhered to stub and coated with a thin gold layer with an accelerating voltage of 30 kV.

### 2.8. Roughness

Film roughness was evaluated with a Mahr CWM 100 confocal microscope (Mahr, Gottingen, Germany). The results were registered after observing 5 different areas and then they were processed with the trial version of Mountain Map software, version 9 (Digital Surf, Lavoisier, France) [24].

### 2.9. FTIR

Agar, alginate and a mixture of these polysaccharides in various proportions with additives such as AA and CaCl_2_ to improve the properties of films were analyzed using FT-IR spectroscopy. The chemical components of the membranes and possible interactions with the additives were recorded using a Nicolet iS20 spectrometer from Thermo Scientific (Karlsruhe, Germany) mounted with an attenuated total reflectance (ATR) accessory and equipped with a diamond crystal. The spectra were registered within the range of 400 cm^−1^ to 4000 cm^−1^ and a detector at 4 cm^−1^. The data were processed with OMNIC software.

### 2.10. Statistical Analysis

All experiments were performed in triplicates. Statistical software SPSS 25.0 (trial version) (IBM, NY, USA) was used to calculate the means and standard deviations for all the parameters investigated. The difference between the data were investigated using an analysis of variance (ANOVA) followed by the Turkey’s HSD test and OriginPro 2019b 9.6.5.169 (Student Version) Copyright © 1991–2019 OriginLab Corporation, Northampton, MA, USA.

## 3. Results

### 3.1. Barrier Properties

#### 3.1.1. Water Content (WC %) and Water Activity (a_w_)

The moisture content increased with the increase in thickness, so the SAA5 (73.10 µm) and control samples (81.3 µm) had the highest moisture content, namely 16.44% and 13.50%, respectively (Table 1).

Due to the fact that all samples have the same matrix, the presence and the amount of the added additives influenced the membranes properties (water content, WVP and WVTR).

Both samples with the addition of AA and those with CaCl_2_ showed a good water activity, much smaller than 0.6, which allows them to be used in the packaging of food products (Table 1). The values of the water activity allowed the appreciation of the packaging safety. Molds, yeasts and bacteria cannot grow at a_w_ values below 0.65 [37].

The membrane formulations with AA presented a_w_ values ranging from 0.38 to 0.51. The membranes with calcium chloride showed smaller a_w_ values (0.39–0.49). The formulations with CaCl_2_ and AA were statistically different (95% confidence level). No significant differences were found between the formulations with AA.

Water content analysis showed that water was not only associated with the film matrix but it was also retained due to the presence of the two additives. This fact was observed in SAA5 and SCa4, in which the water content exceeded the control sample (Table 1). The difference in moisture can be attributed to the hygroscopic activity of AA and CaCl_2_. Moreover, AA have four hydrogen bonds (donor sites), making it a strong component for water absorption [38].

#### 3.1.2. Water Vapor Permeability (WVP) and Water Vapor Transmission Rate (WVTr)

WVP indicates the ability of the membranes to prevent moisture transfer [39]. A large WVP negatively influences the quality of food, resulting in shorter shelf-lives and a possible increase in the amount of waste [16].

In the samples with AA addition, the lowest values for WVP were recorded for SAA4 and SAA1. Regarding the samples with CaCl_2_ addition, lower values for WVP and WVTr were recorded for SCa3 and SCa4 (significant difference at 95% confidence level).

The WVP values presented small variation as the amount of AA was added was between 0.20 and 0.59. The variation was smaller for CaCl_2_ membranes, between 0.15 to 0.39, due to the compact network made of calcium salt, as shown in Table 1. All the differences found were significant (*p* < 0.05).

Regarding the WVTr the lowest values were recorded in the samples SAA4, SAA1, SCa3 and Sca4 (Table 1).

#### 3.1.3. Oxygen Barrier Property Measurement (OP)

Exposure to oxygen causes lipid degradation and changes in color and taste of the product. Oxygen barrier properties were assessed by PV. Membranes made with the addition of AA and those with CaCl_2_ have shown a very good barrier to oxygen, with a PV = 0 meqO_2_/g. Only, the control sample had PV = 0.007 meqO_2_/g. The obtained results showed that all the membranes with additives formed a very good barrier to the penetration of atmospheric oxygen, protecting the products from unwanted oxidation reactions. Membranes had a good barrier to the penetration of oxygen when their humidity not exceeded 20% [16]. Above this value, the mobility of the polymer chains increases, which facilitates the transport of oxygen through the membrane. Except for the control sample, all membranes analyzed had humidity between 9.23 and 16.44%, which explains their very good barrier effect and lack of oxygen transfer.

#### 3.1.4. Grease Permeability Measurement

After weighing the filter paper that was in contact with the membrane that closed the oil bottle, it was found that the weight remained constant, therefore none of the membranes allowed the passage of the oily phase. Similar results were obtained by Chang et al. (2019) [13] in the characterization of the films made.

### 3.2. Film Thickness

The thickness of the membranes (Table 1), for the same matrix, was influenced by the amount of additive added. The thickness values of the membranes with AA addition varied were between 57.3 µm and 73.1 µm, and between 52.75 µm and 71.45 µm for those with CaCl_2_ (Table 1). These values can be explained by the interruption of the connection between biopolymers caused by the two additives. Compared to the control sample, the lowest thickness was recorded in SAA3 and SCa4, while the highest thickness was in the case of SAA5 and SCa2. As can be seen, CaCl_2_ membranes are thinner than AA membranes. The thickness of the CaCl_2_ membranes was significantly different (*p* < 0.05) from those with AA, possibly due to the differences in structure, chemical composition, hydrophobicity, and properties of the two constituents.

### 3.3. Mechanical Properties of Membranes

The mechanical properties were influenced by the presence of AA and CaCl_2_ (Table 2).

Compared to the control sample, those with AA, presented the highest values (56.43 MPa) for tensile strength (TS). Moreover, the highest value of TS in the added samples was recorded in SAA1 (53.67 MPa) with 0.1 g. The results showed that as the amount of AA increased, the TS and elongation at break (E) values decreased, due to the good compatibility of AA with the biopolymers, which allowed it to interfere between their molecules through hydrogen bonds [40]. Decreasing in the TS value showed a decrease in the number of hydrogen bonds between the biopolymer molecules and the formation of new bonds with the additive. The same results were obtained by Lipka et al. [41].

The SAA1 sample with the lowest amount of AA added showed the best mechanical properties. The elasticity decreased with the increase in the amount of AA added, being the highest in the SAA1 sample. There was a decrease in hardness (H) by almost of 35.7% from the SAA1 sample to the last SAA5 sample as the amount of added AA increased. AA is considered a weak acid, that induces plasticization effect, which caused a reduction in tensile strength TS and in elongation at break E%.

The effect of CaCl_2_ addition on the mechanical properties of membranes was strong due to the network that is formed between calcium ions and carboxyl groups of alginate. Thus, the sample SCa2, with 0.02 g CaCl_2_ added, presented good values for TS, and E. SCa4 sample with 0.08 g CaCl_2_ had the highest value for TS, an average value for E, but a low value for hardness. A similar variation in the two parameters was mentioned by Santana et al. (2013) [12]. There was a decrease in hardness (H) as the amount of added Ca^2+^ increased by almost of 40% from the first to the last sample due to the presence of Ca^2+^ cation in the newly formed network of biopolymers. The effects of using CaCl_2_ crosslinker in similar amounts to those used on the obtained membranes were highlighted by Farhad Garavand et. al, (2017) [42]. By increasing the amount of CaCl_2_ in the membranes, the tensile strength TS effect was also increased, as also mentioned by Mariana Altenhofen da Silva et al. (2009) [43]. Sodium alginate in the presence of higher amounts of calcium ions formed a stronger network, which decreased the value of E%, also observed by E.M. Zactiti et al. (2009) [44].

The TS values obtained for all the membranes fell within the limits imposed in the standard (ASTM D 638 method [45]) for the polyethylene foils. According to the standard the value must be of min. 29 MPa. The limits of the allowed values for TS (64.42 ± 3.78 MPa) and E% (36.26 ± 7.43 %) for LDPE were similar to those presented by Jung H. Hun and Aristippos Cennadios (2005) [46].

### 3.4. Optical Properties of the Membranes

Color is the first attribute of a package that can be correlated with the acceptance or rejection of the food by the consumer [47]. The influence of the addition of AA or CaCl_2_ on the optical properties of the membranes was evaluated by the value of the sizes in Table 3.

The addition of AA decreased the lightness of the membranes, an increased the b* value and therefore an increased in the yellowness of the biopolymer-AA membranes (this can also be seen in Figure 1), due to the oxidation of AA to dehydroascorbic acid, according to Kowalczyk, Dariusz (2016) [48]. Compared to the control sample, the opacity of the membranes increased, but it decreased with the increase in the amount of AA added (Figure 1). Similar results were obtained by Al Luqman Abdul Halim et al., 2018 [20]. The darker green color opened up (from SAA5 to SAA1) with the addition of AA compared to the control sample. The yellow color of the membranes was accentuated as the amount of AA increased.

ΔE increased significantly (*p* < 0.05) in agreement with the higher a and b (Table 3).

In the case of samples with the addition of CaCl_2_, sizes L*, a* and b* (Table 3, Figure 2) show a very small variation, the color of all the samples ranging from very light red to very light yellow.

The total color differences (ΔE) were determined to see if the addition of the two substances influences the total overall color of the membranes. A higher value than 2.00 confirms a significant influence on color, the difference (Δ*E*) increased significantly in agreement with the higher a* and b* [32]. The largest color variation was recorded in samples with AA, from 6.6 to 18.23 as the amount of AA added increased. In the case of CaCl_2_ addition, the color variation was much smaller from 1.11 to 0.02 and it decreased with the amount added from 1.0 to 0.02 due to the stability of the calcium compound.

Membrane transparency positively influences the attractiveness of consumers who have the opportunity to view and analyze what they buy. For food products, transparency is important, but also the protection of the products against light radiation, which can negatively influence the quality by changing the color, by oxidizing lipids, especially in the case of food products. As shown by the values of opacity and transparency in Table 3, as well as by the variations in these sizes between 200–780 nm in the visible domain, the obtained membranes have both transparency and opacity necessary for protection against UV radiation.

All the films (Figure 3a,b) showed a very good UV absorption. Light UV absorbance was better than polymeric materials such as low-density (LDPE) polyethylene (Figure 3c). The results obtained from the membrane analysis are in accordance with the data specified by Melina Dick et al. 2015 [14]. The UV light corresponds to a 200–280 nm region, and as shown from Figure 3, the light absorbance (%) was very low for all the samples in this range, which implies that the membranes with AA and CaCl_2_ added have the ability to protect against UV radiation due to their very good barrier capability.

For the samples with CaCl_2,_ the best absorption of UV radiation was between 200 and 400 nm, so the best protection was registered at SCa1 (Figure 3b), which was the sample with the smallest amount of CaCl_2_ added. All samples (Figure 3b) with CaCl_2_ addition had a similar absorption in the visible field, which provides consumers with high visual access. In the case of samples (Figure 3a) with added AA, the protection against UV radiation increased with the amount of AA from SAA1 to SAA5. Both AA and CaCl_2_ samples showed better values for UV and VIS absorption than the control sample.

The absorption properties of light radiation by the PE film currently used as a packaging material of the food products are shown in Figure 3a,b. The analysis of the absorption spectra of PE film showed much lower values of UV absorption than those of the obtained membranes, therefore, a lower protection of the food against the influence of light radiation.

### 3.5. FTIR—ATR Analysis

The spectrum of agar exhibited various characteristic peaks in the range from 3354 cm^−1^ to 771 cm^−1^. The characteristic broad absorption band at about 3354 cm^−1^ indicated stretching of hydroxyl (O-H) groups [43,44,45,46].

All membranes (Figure 4) presented wave numbers in four different spectral zones: 3500–3200 cm^−1^, 2900–2930 cm^−1^ and 1000–1030 cm^−1^, which can be assigned to bond stretching of the groups OH, CH and COC, respectively, present in the alginate structure. Furthermore, peaks at the wave numbers of 1600–1605 cm^−1^ and 1409–1412 cm^−1^ can be assigned to asymmetrical and symmetrical stretching of the COO- bond, respectively [47].

The FTIR spectrum of pure glycerol showed five typical absorption bands located at 800 up to 1150 cm^−1^, corresponding to the vibrations of C–C and C–O linkages. Three broad bands at 850, 925, and 995 cm^−1^ corresponded to the vibration of the skeleton C–C; the peak at 1045 cm^−1^ was associated to the stretching of the C–O linkage, and the bond at 1117 cm^−1^ was corresponded to the stretching of C–O [43].

The peak absorption of membranes with added calcium chloride was lower than the peak corresponding to the control sample due to the network created between the calcium ion and alginate, which led to a decrease in the interaction between the two biopolymers and plasticizer. The number of COO– groups decreased with the formation of hydrogen bonds with the OH groups of agar and glycerol, to which was added the binding of Ca^2+^ ion in case of the addition of calcium chloride.

Compared to the standard sample, the bands corresponding to the Ca^2+^ samples have been widened and shifted to lower wave numbers due to the decrease in the number of intermolecular hydrogen bonds. The band corresponding to the wavelength of 1644 cm^−1^ was attributed to the vibrations of the HO group and the 1619 cm^−1^ wavelength band was assigned to carboxyl groups. The carboxyl groups were displaced from 1619 cm^−1^ to 1606 cm^−1^ and 1612 cm^−1^, respectively, indicating the hydrogen bond between the HO group of the agar and the carbonyl group of the AA. In the presence of Ca^2+^ ion, alginate formed a chelate with the carboxyl group, a compound that inhibits the interaction of alginate, agar and glycerol. Furthermore, the characteristic broad absorption bands of AA was described by Domínguez-Martínez, et al., 2014 [49].

### 3.6. Microstructure of Biodegradable Membranes

Scanning electron microscopy was conducted to visualize the differences in the membrane morphology at AA and CaCl_2_ addition to a matrix containing alginate, agar and glycerol in the established ratio.

SEM micrography of the samples confirmed the existence of crystal formation (see Figure 5d) in the presence of CaCl_2_. The surface of the sample was more uniform and smoother than that of the control sample as can be seen in Figure 5c,d. Alginate reacted with many divalent cations, but it led to the formation of the most stable and uniform networks with the Ca^2+^ ion [38]. The sample with AA, Figure 5a,b had a surface as well as a less uniform section than the control sample, Figure 5e,f, due to the influence of the additive on the interaction between the components. Electron micrographs of the cross-section of all membranes showed a compact and homogenous structure without pores, which indicates complete miscibility of the components in the mixture.

### 3.7. Determination of Roughness

The roughness of the samples was determined on both sides of the membranes. At the surface in contact with the silicone foil the roughness was lower, the water from the formed hydrogel must cross the thickness of the film to reach the surface, the lower speed, the thicker the membrane is. On the free surface the roughness was higher due to the rapid evaporation of water from the membrane surface in contact with the surrounding air.

As can be seen from Figure 6A the total roughness of the air-side surface of the SCa4 sample was 33.813 µm, and it was higher than that of the surface in contact with the silicone foil, which was 30.117 µm (Figure 6B). Furthermore, the roughness of the free surface was uneven due to the rapid evaporation of the water taken up by the air, with which it was in contact, while the roughness of the surface in contact with the silicone foil was uniform, the water molecules crossing the thickness of the film were at a low speed. On both sides the roughness of the samples was very good due to a small thickness of the foil, of 52.75 µm (the thinnest of the membranes with calcium chloride) and the low density of 1.5 g/cm^3^. The low values of roughness and the uniformity of the surface are explained by the network formed by calcium ions with sodium alginate, and the small difference between the roughness of the two faces can be attributed to the stability of the formed network. The total roughness of the SAA1 sample was 49.865 µm, (Figure 6C) and it was higher than that of the surface in contact with the silicone foil, which was 34.987 µm (Figure 6D).

The large difference between the roughness of the two surfaces of the control sample was due to the rapid evaporation of unbound water from the outer face of the membrane (Figure 6E,F).

When packaging the food products, the surface of the membrane with higher roughness will be placed inside, and the one with lower roughness outside so as to reflect light and create a favorable image of the product.

Physicochemical analyses, mechanical strength, structure, roughness, optical properties, antiviral, antibacterial and antioxidant capacity of the membranes based on biopolymers with the addition of ascorbic acid or calcium chloride demonstrated the quality of these materials and the possibility of replacing the classic ones made of polyethylene in the packaging of some food products.

## 4. Conclusions

To improve the properties of the biodegradable membranes in the mixture of biopolymers (alginate, agar) and plasticizer (glycerol) was added AA or CaCl_2_.

The aim of the research was to obtain resistant and elastic materials, which would allow the packaging, transport and storage of food products and remain in good condition. Other requirements were: protection against UV radiation; sufficient transparency in the VIS field; uniform pore-free structure and very good water, oxygen, and fat barrier properties.

So far no membranes have been proposed to cover the trays with sliced food. The work can be followed by other studies with the same purpose, the improvement in chemical composition, in order to replace the large quantities of PE accumulated due to a consumer preference for smaller portions.

The two types of membranes obtained by adding both AA and CaCl_2_ in filmogenic solutions of biopolymers have a uniform structure, free of pores or cracks, and the strength and the elasticity necessary for packaging, storage and transport of food. All the membranes obtained offered good protection against the action of UV radiation and sufficient transparency in the VIS field for the visualization of the products. Their good barrier properties in contact with the external environment and water activity ensure microbiological safety in the consumption of products.

All samples presented lower thickness, but superior mechanical strength to the control samples, which is important for saving packaging material.

Improvement of membrane properties by the addition of ascorbic acid and calcium chloride and the reconfiguration of the bonds between the components of the mixture, has been demonstrated and confirmed by the FTIR data. Physico-chemical analyses, mechanical strength, structure, roughness, optical properties, antiviral, antibacterial and antioxidant capacity of the membranes based on biopolymers with the addition of ascorbic acid and calcium chloride demonstrated the quality of these materials and the possibility of replacing the classic ones made of polyethylene or polypropylene.

## Figures and Tables

**Figure 1 membranes-12-00576-f001:**
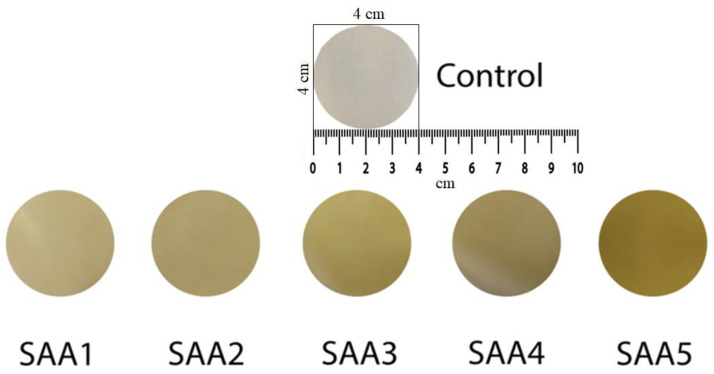
Color comparison between samples with ascorbic acid (SAA) addition and control sample. SAA1-5 concentration from 0.10 to 0.45 g.

**Figure 2 membranes-12-00576-f002:**
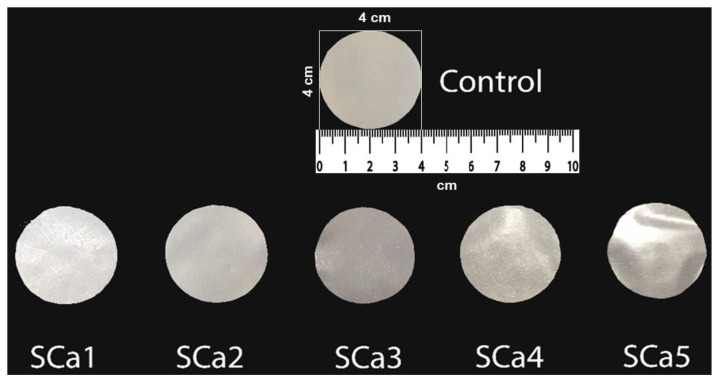
Variation in color between the samples with calcium chloride (SCa) addition and the control sample. SCa1-5 concentration from 0.01 to 0.1 g.

**Figure 3 membranes-12-00576-f003:**
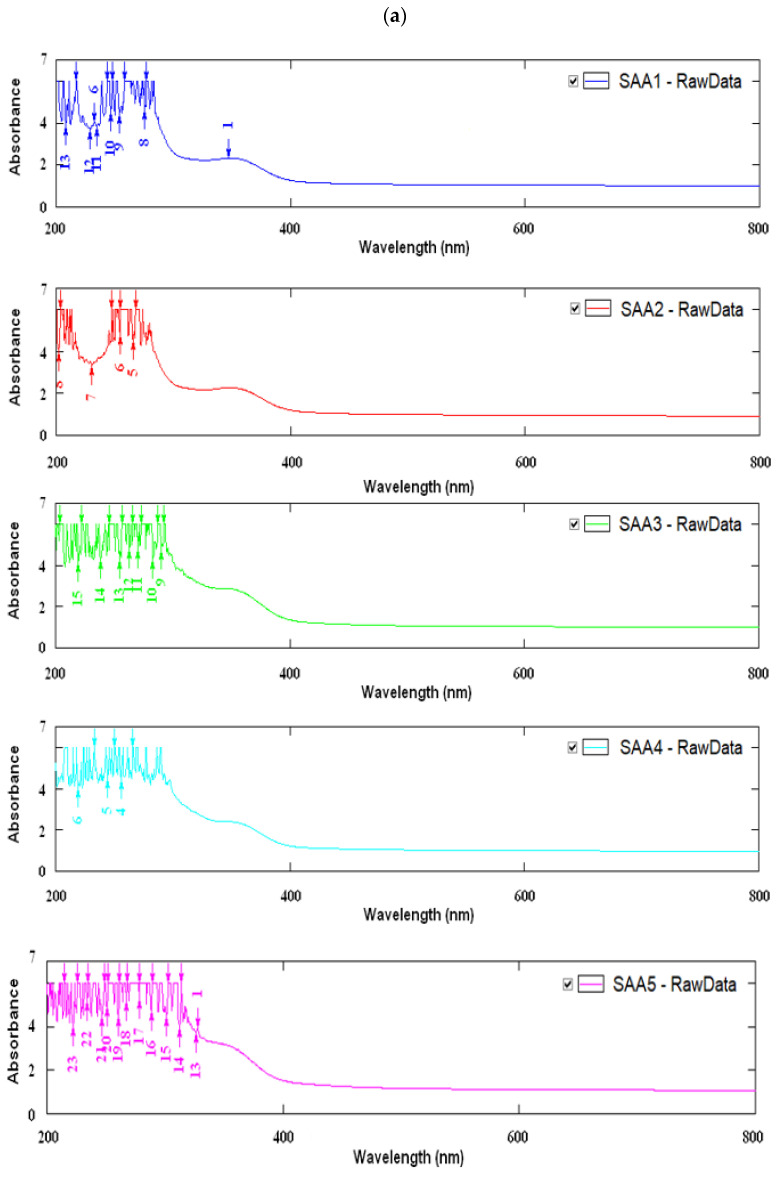
Stacked absorption spectra between 200 and 800 nm of the membranes with AA (**a**), calcium chloride (**b**), control sample and PE membrane (**c**).

**Figure 4 membranes-12-00576-f004:**
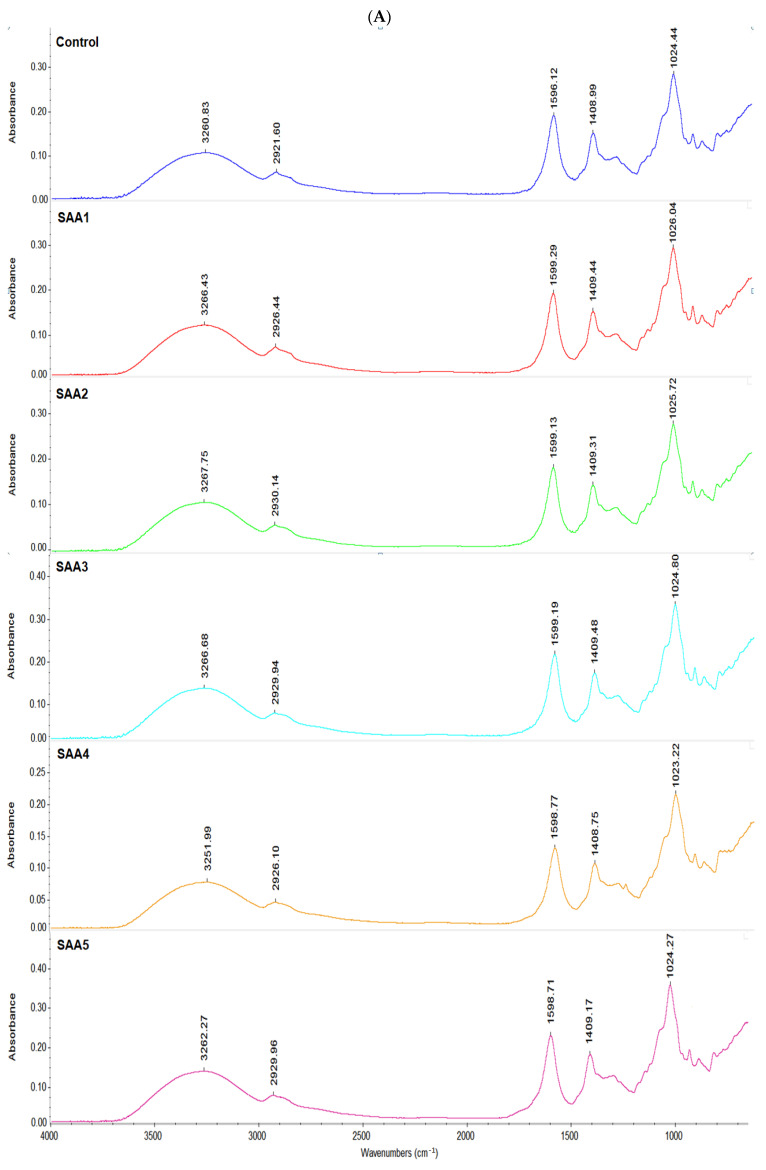
FTIR-ATR spectrograms: (**A**)—stacked absorption spectra of the membranes with AA addition and control sample; (**B**)—stacked absorption spectra of the membranes with CaCl_2_ addition and spectra of the control sample; (**C**)—stacked absorption spectra of the individual compounds.

**Figure 5 membranes-12-00576-f005:**
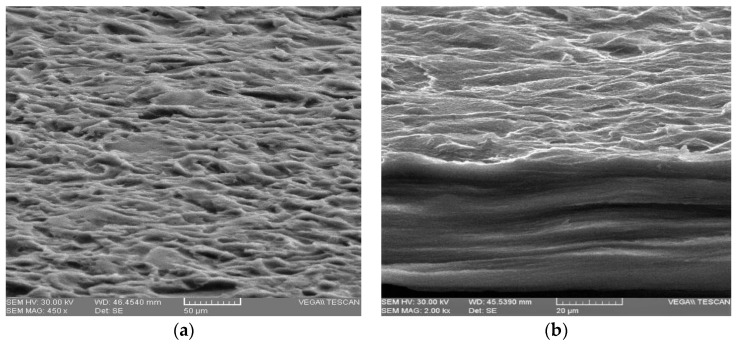
FE-SEM images showing surface morphologies of membranes and cross-sectional images: (**a**,**b**)–SAA1; (**c**,**d**)–SCa4; (**e**,**f**)–control sample-without additives.

**Figure 6 membranes-12-00576-f006:**
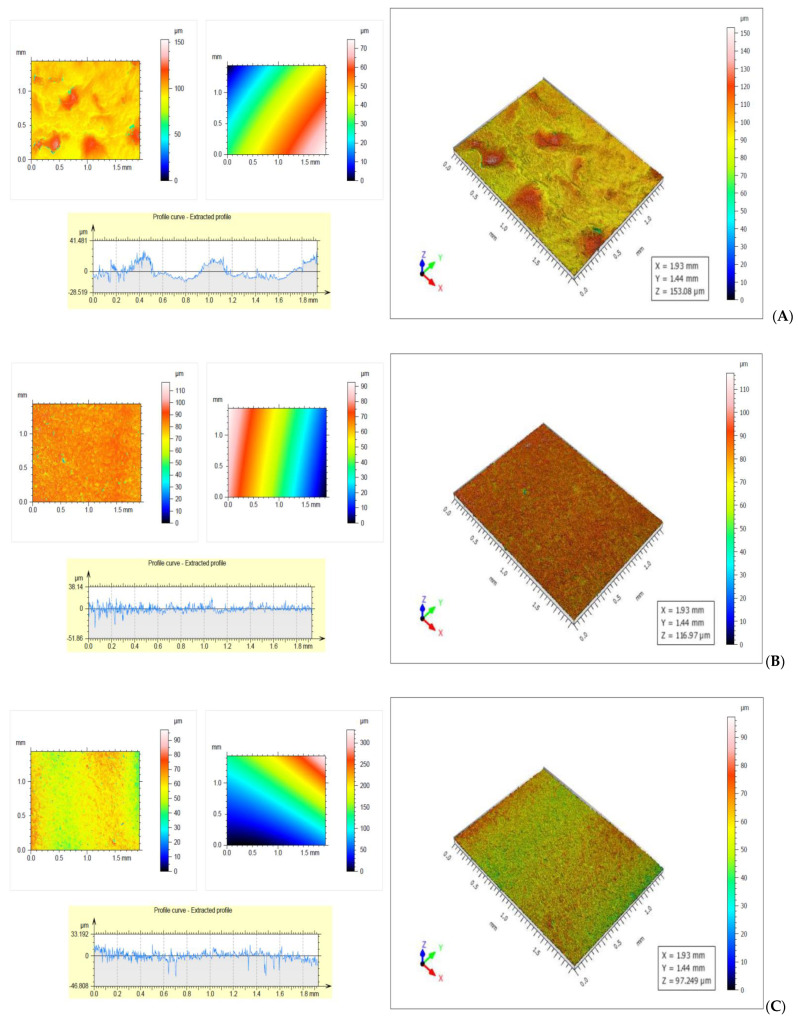
(**A**)—Rough surface of SCa4 sample—Rt = 33.813 µm; (**B**)—Glossy surface of SCa4 sample—Rt = 30.117 µm; (**C**)—Rough surface of SAA1 sample—Rt = 49.865 μm; (**D**)—Glossy surface of SAA1 sample—Rt = 34.987 µm; (**E**)—Rough surface of Control sample—Rt = 70.335 µm; (**F**)—Glossy surface of control sample—Rt = 38.357 µm.

**Table 1 membranes-12-00576-t001:** Barrier properties of membranes with a fixed mixture of alginate and agar and added with an increasing amount of AA and CaCl_2_.

Sample	Thickness (µm)	Water Content (%)	Water Vapor Permeability (g mm/kPa h m^2^)	Water Vapor Transmission Rate (g/m^2^ h)	a_w_
SAA1	65.40 ± 5.10 ^d^	18.68 ^a^	0.22 ± 0.001 ^e,f,g^	10.77 ± 0.01 ^e^	0.51 ± 0.002 ^a^
SAA2	66.25 ± 4.68 ^c,d^	12.28 ^d^	0.59 ± 0.001 ^b^	12.43 ± 0.01 ^d^	0.47 ± 0.002 ^b^
SAA3	57.30 ± 4.32 ^g^	11.05 ^f^	0.30 ± 0.002 ^c,d,e^	16.32 ± 0.01 ^b^	0.48 ± 0.010 ^b^
SAA4	62.70 ± 3.33 ^e^	9.75 ^h^	0.20 ± 0.001 ^f,g^	9.98 ± 0.01 ^f^	0.38 ± 0.003 ^f^
SAA5	73.10 ± 4.81 ^b^	16.44 ^a^	0.27 ± 0.002 ^d,e,f^	16.3 ± 0.01 ^b^	0.48 ± 0.020 ^a,b^
SCa1	58.70 ± 1.76 ^f,g^	11.49 ^e^	0.35 ± 0.001 ^c,d^	18.80 ± 0.02 ^a^	0.42 ± 0.010 ^d^
SCa2	71.45 ± 3.55 ^b^	11.51 ^g^	0.39 ± 0.001 ^c^	16.39 ± 0.01 ^b^	0.44 ± 0.030 ^c^
SCa3	67.70 ± 3.65 ^c^	9.23 ^i^	0.18 ± 0.001 ^a^	8.79 ± 0.01 ^h^	0.39 ± 0.010 ^e^
SCa4	52.75 ± 4.24 ^h^	13.87 ^b^	0.15 ± 0.001 ^g^	9.23 ± 0.01 ^g^	0.49 ± 0.030 ^a^
SCa5	60.30 ± 3.75 ^f^	10.51 ^a^	0.23 ± 0.002 ^e,f,g^	12.12 ± 0.01 ^d^	0.44 ± 0.01 ^c^
Control	81.30 ± 4.94 ^a^	13.50 ^c^	0.35 ± 0.001 ^c,d^	13.44 ± 0.01 ^c^	0.48 ± 0.030 ^a,b^

SAA1-SAA5, samples with 0.10 to 0.45 g AA addition; SCa1-SCa5, samples with 0.01 to 0.1 g CaCl_2_ addition. Different letters are significantly different. The analysis was performed in triplicates. Values are presented as mean ± standard deviation. Values followed by different superscript letters (a, b, c, d, e, f, g, h, i) are statistically different at 95% confidence level.

**Table 2 membranes-12-00576-t002:** Mechanical properties of the membranes with the addition of AA and CaCl_2_.

Samples	Density (g/cm^3^)	Tensile Strength/TS (MPa)	Hardness (H)	Elongation at Break/E (%)
SAA1	1.61 ± 0.020 ^b^	53.67 ± 3.66 ^c^	322.50 ± 10.66 ^d,e^	12.69 ± 0.17 ^a^
SAA2	1.30 ± 0.010 ^h^	48.46 ± 3.33 ^d^	271.30 ± 21.38 ^h^	4.55 ± 3.11 ^i^
SAA3	1.85 ± 0.001 ^a^	34.26 ± 1.71 ^f^	398.40 ± 15.23 ^b^	6.63 ± 3.52 ^g^
SAA4	1.43 ± 0.003 ^f^	52.85 ± 4.02 ^c^	314.20 ± 10.89 ^e^	11.96 ± 0.30 ^b^
SAA5	1.31 ± 0.002 ^g,h^	40.77 ± 0.40 ^e^	252.00 ± 35.20 ^i^	8.43 ± 2.42 ^e^
SCa1	1.35 ± 0.001 ^g^	22.88 ± 1.03 ^h^	288.20 ± 14.64 ^g^	2.56 ± 1.18 ^j^
SCa2	1.54 ± 0.001 ^c^	53.84 ± 3.98 ^c^	326.66 ± 24.01 ^d^	11.95 ± 0.03 ^b^
SCa3	1.49 ± 0.003 ^d,e^	56.41 ± 2.44 ^b^	297.00 ± 10.32 ^f^	8.07 ± 1.03 ^f^
SCa4	1.50 ± 0.050 ^c,d^	67.34 ± 2.45 ^a^	280.30 ± 10.72 ^g^	9.44 ± 3.18 ^d^
SCa5	1.45 ± 0.010 ^e,f^	26.93 ± 2.29 ^g^	422.90 ± 11.64 ^a^	5.60 ± 1.15 ^h^
Control	1.32 ± 0.001 ^g,h^	56.43 ± 2.80 ^b^	337.50 ± 9.18 ^c^	10.41 ± 0.08 ^c^

SAA1 to SAA5, samples with AA addition; SCa1 to SCa5, samples with CaCl_2_ addition. Values are presented as mean ± standard deviation. Values followed by different superscript letters (a, b, c, d, e, f, g, h, i, j) are statistically different at 95% confidence level.

**Table 3 membranes-12-00576-t003:** Variation in the optical properties of the membranes with different additions of AA and CaCl_2_.

Samples	L*	a*	b*	ΔE	Opacity	Light Transmission	Transparency
SAA1	86.75 ± 0.04 ^c^	−0.19 ± 0.03 ^c^	12.70 ± 0.08 ^d^	6.60 ^d^	18.04 ± 0.001 ^c^	6.61 ± 0.004 ^g^	2.00 ± 0.04 ^g^
SAA2	85.57 ± 0.16 ^c^	−0.49 ± 0.05 ^d^	15.06 ± 0.14 ^c^	9.17 ^c^	14.53 ± 0.001 ^f^	10.88 ± 0.010 ^c^	2.22 ± 0.08 ^c,d^
SAA3	85.79 ± 0.01 ^c^	−1.20 ± 0.01 ^h^	17.70 ± 0.05 ^b^	11.31 ^b^	17.38 ± 0.001 ^d^	10.08 ± 0.002 ^e^	2.25 ± 0.02 ^b,c^
SAA4	87.78 ± 0.01 ^b,c^	−0.66 ± 0.04 ^e^	12.68 ± 0.04 ^d^	5.96 ^e^	15.39 ± 0.001 ^e^	10.87 ± 0.004 ^c^	2.24 ± 0.01 ^b,c^
SAA5	81.76 ± 0.12 ^d^	−0.95 ± 0.02 ^b^	23.39 ± 0.30 ^a^	18.23 ^a^	13.20 ± 0.001 ^h^	10.86 ± 0.010 ^c^	2.17 ± 0.01 ^d^
SCa1	91.30 ± 0.19 ^a^	−0.90 ± 0.01 ^g^	6.90 ± 0.050 ^g^	0.82 ^g^	19.76 ± 0.001 ^a^	7.41 ± 0.010 ^f^	2.10 ± 0.06 ^e,f^
SCa2	90.85 ± 0.14 ^a^	−0.98 ± 0.01 ^a^	7.58 ± 0.004 ^f^	0.02 ^h^	13.59 ± 0.001 ^g,h^	10.69 ± 0.010 ^c,d^	2.17 ± 0.08 ^d^
SCa3	91.34 ± 0.11 ^a^	−0.90 ± 0.06 ^g^	7.02 ± 0.020 ^g^	0.74 ^g^	13.90 ± 0.001 ^g^	11.56 ± 0.003 ^b^	2.23 ± 0.01 ^b^
SCa4	90.40 ± 0.27 ^a^	−0.81 ± 0.01 ^f^	8.59 ± 0.030 ^e^	1.11 ^f^	18.80 ± 0.001 ^b^	10.23 ± 0.010 ^d,e^	2.29 ± 0.02 ^b^
SCa5	90.07 ± 1.18 ^a,b^	−0.88 ± 0.01 ^g^	6.91 ± 0.010 ^g^	1.05 ^f^	14.48 ± 0.001 ^f^	11.47 ± 0.020 ^c,d,e^	2.84 ± 0.04 ^a^
Control	90.86 ± 0.02 ^a^	−0.97 ± 0.02 ^a,b^	7.59 ± 0.030 ^f^	-	12.06 ± 0.001 ^i^	10.47 ± 0.001 ^c,d,e^	2.11 ± 0.06 ^e,f^

SAA1, sample with 0.10 g AA addition; SAA2, sample with 0.20 g AA addition; SAA3, sample with 0.25 g AA addition; SAA4, sample with 0.30 g AA addition; SAA5, sample with 0.45 g AA addition; SCa1, sample with 0.01 g CaCl_2_ addition; SCa2, sample with 0.02 g CaCl_2_ addition, SCa3, sample with 0.04 g CaCl_2_ addition; SCa4, sample with 0.08 g CaCl_2_ addition; SCa5, sample with 0.1 g CaCl_2_ addition; ΔE, degree of total color difference from the standard color plate. Values followed by different superscript letters (a, b, c, d, e, f, g, h, i) are statistically different at 95% confidence level.

## Data Availability

Not applicable.

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
