# Peer review of "Development of New Biodegradable Agar-Alginate Membranes for Food Packaging"

_membranes, 2022, doi:10.3390/membranes12060576_

Round 1

Reviewer 1 Report

The research paper deals with the development of an active packaging material based on alginate, agar and glycerol as plasticizer with potential to cover trays with dry salami slices. The topic is current and very interesting results are showed with a wide physical-mechanical characterization of the developed material. The manuscript can be improved and some aspects to be considered are presented below:

-In section 3.1.3, it should be presented results for all samples with more details if it is possible to better support discussions. Indicate in the experimental part meaning of IP.

-The level of ascorbic acid and calcium chloride could be presented in terms of concentration, e.g. respect to the total mass or polymer mass.

-In the abstract and introduction is not clear if the ascorbic acid and calcium chloride were added together or if their effect were studied separately. Please, rewritten or rephrase where be necessary.

-Line 35: Rephrase to clear the meaning of “...due to the accumulation of its properties”

-Line 55: Rephrase to clear the meaning of “…hydrocolloids class…”.

-Line 64-65: Please, include references that show “a low amount of glycerol….will reduce the water vapor permeation”

-Line 69: ¿What do authors mean with “a significant effect”?. Improvement or deterioration? Please, give details to clarify.

- The aim of the work could be better presented in the introduction describing it with more details and indicate what is the differences with the background (references). For example, have been AA and CaCl2 used before in alginate, agar or similar polymers?

-Line 339-340: Include the references where AA reduced the elongation at break in similar polymers. The plasticizing effect became in a reduction of the elongation at break?

-Please, the discussion will be improved and indicate some details about parameters with values without a clear trend. For example, in hardness, where highlighted that SAA4 has the highest value but them decreased. Similar with TS in SCa4. Please, include more discussion about that differences and/or indicated the case where is there is only a trend.

-Line 348-349: The sentence “The sample SCa2…had the highest value for TS”. This is not coherent with the results in the table.

-Line 359-364: The paragraph should be rewritten to a better understanding and improve the comparisons of the results obtained in the developed material and polyethylene.

-Conclusions can be improved highlighting and extending the best characteristics of the developed biodegradable material in this work and the relationship between structure-properties. Conclusions are very

Improve the English grammar in the manuscript for a better and simple understanding, for example, in Line 326, change “the acid ascorbic added samples” to “samples with acid ascorbic” or “samples with AA”. Consider to use the abbreviations that were indicated before.

Reviewer 2 Report

Introduction

>>>>> The first paragraph of the introduction is overly general and is not specifically related to the content of the study. The introduction should be revised for a better focus of the problem studied. The main part is one very long paragraph that should be separated into concepts. The aims are also very unclear and do not match with the abstract.

Methods

>>>>> Section 2.1: How much water was used? The detail of this method is generally unclear and not logically presented.

>>>>> Section 2.10: Triplicate measurements are fine except for the mechanical properties. At least 5 measurements are needed for any meaningful statistics.

Results and Discussion

>>>>> Table 1: at no point in the methods are the sample abbreviations given, yet SAA1-5 and SCa1-5 are given in this table. The reader has to assume which samples are presented.

>>>>> Line 272: citation for this statement?

>>>>> Line 359: how is this relevant for a biopolymer film?

>>>>> Why Figure 3a then 3g?, It should be logically 3a and 3b. Wavelength is misspelled on both x-axes, and Figure 3a is extremely saturated below about 300 nm.

>>>>> Line 437: where is the glycerol spectrum?

>>>>> It is difficult to confirm the discussion of the FTIR data. Drop-lines would make it easier to show shifts in key wavelengths. The color choice is also very poor.

>>>>> In Figure 7 you show some packaged salami but what was the purpose? It wasn’t mentioned in the methods and there are no analyses performed on the salami samples.

Other comments

>>>>> I do not agree that you have demonstrated the films could “successfully replace the synthetic petroleum-based packaging materials” line 26.

>>>>> The term “edible” is used frequently but in reality, there are very few truly edible films. Particularly overwrap for meat products. This term should be removed throughout.

Round 2

Reviewer 1 Report

The manuscript discussions were not sufficiently improved. The made changes were weak. The aim of the work in the introduction section was not sufficiently improved. The results of mechanical properties were not well revised and requested explanations about the not clear trends in mechanical results were not included. Not clear trends in results can be attributed to the number of replicates used for the mechanical tests. Three replicates are not sufficient for a tensile/mechanical test. 

On the other hand, the composition of the samples in the experimental section is not well presented yet. The style for describing samples is confusing. 

If experiments about antiviral, antibacterial and antioxidant capacity were not carried out in the work, ¿Why the authors claim in the conclusion, "antiviral, antibacterial and antioxidant capacity of the membranes based on biopolymers with addition of acid ascorbic and calcium chloride demonstrated the quality of these materials"?. 

Reviewer 2 Report

The manuscript is improved but there are still some issues.

The introduction is better but the large paragraph should be split as I previously suggested (unless the track-changes is not showing a split). There was no explicit response to this comment.

Figure 3b still shows unacceptable saturation, which seems to be due to the ascorbic acid. This is not discussed but may be associated with the color observed.

In your response, you included some good information about the down-sizing of packaged foods as justification for including the salami. None of this has been translated to the manuscript, consider adding this to the introduction. 

Moreover, the inclusion of Figure 7 is still overly qualitative and does not add any value to the manuscript. This could have been any food and you have not demonstrated any adequate use for this film (i.e., no tests on the salami) although you discuss it indirectly throughout.

Since you have not demonstrated any real effect of the film to package the dry salami, it’s inclusion (including the title), could be misleading to any readers/researchers who would otherwise assume this manuscript is related to dry salami preservation. If you focus on just the film development and testing, the manuscript would be a better contribution. The dry salami is irrelevant.

Round 3

Reviewer 1 Report

The authors improved the manuscript. 

ASTM D882 allows a reduced number of samples under some considerations Section 9 (see below). Nonetheless, in the first revision, it was requested to improve discussions on mechanical properties about unclear trends in mechanical parameters and mention possible factors influencing them.

9. Number of Test Specimens
9.1 In the case of isotropic materials, at least five specimens
shall be tested from each sample.
9.2 In the case of anisotropic materials, at least ten
specimens, five normal and five parallel with the principal axis
of anisotropy, shall be tested from each sample.
9.3 (Optional) It is acceptable to test a reduced number of
test specimens:
(1) No less than three test specimens shall be tested.
(2) No less than six test specimens in the case of anisotropic
materials, three normal and three parallel with the principle
axis of anisotropy, shall be tested.
(3) Allowed for in-line quality control sampling.
(4) Allowed for samples not sufficient in size to provide a
minimum of five test specimens (10 test specimens for anisotropic
materials).
(5) Standard deviation is not to be calculated or reported
due to the reduced number of data points.

Author Response

This manuscript is a resubmission of an earlier submission. The following is a list of the peer review reports and author responses from that submission.

Round 1

Reviewer 2 Report

Biopolymer-based active packaging materials, a possibility to  reduce the use of plastics by the food industry.

The positive aspect of manuscript:

The advantages of environmentally friendly and even edible coatings for food packaging are given clearly, so the general necessity of research is well justified.The research has pottential impact in the field of development of food packaging materials.

However, the results could be presented in more correctly. More details please see in Comments below.

Comments:

  1. Language must be edited - thera are a lot of  mix of past and present sentences.
  2. Section 2.1. what was the concentration of polymer solution for film casting? What was the planned and obtained grammage (g/m2) for films? It should be taken into account, if films will be compared. 
  3.  Section 2.4.2. – what was the RH in the dessicator?
  4. Section 2.4.3. was the experiment only 3min long? Do authors consider that it is enough to conclude about oxygen permeation trough mambranes? Why didnt authors choose longer period of time for oxygen transfer? Was it new method or any standard or published method used as reference?
  5. Line 187 – film conditioning is mentioned first time in themanuscript . Please explain, why condioning was not used before other testing methods? How did the authors make sure that all samples of films have  the same moisture conditions? Moisture content can have significant effect on film barrier properties and results without preliminary conditioning are not reliable.
  6. Section 2.7. Please give more details about film preparation for SEM. What was the euipment used and parameters of procedure?
  7. Section 2.9. Were FTIR spectras obtained for dry polysachcarides and additives or for final films? It is not clear from description of method. If it for dry substances, what was the moisture content? How did authors evaluate the interactions of components, if they were not in the final film form?
  8. Figure 1 – SAA1 and SAA2 seems similar compositions. Please explain if so or give correct numbers in the graph.
  9. Figure 2 – it would be better to rearrange the order of compositions in the graph according to progressivly increasing content of additive.
  10. Line 268-275. Please explain more measured water activity, the meaning of obtained numbers. What is the actual functionality of this measurement/calculations?
  11. Films can be correctly compared regarding their properties, if have the same moisture content, which can be obtained by conditioning. It is necesarry to explain, why barrier properties were compared for for non-conditioned films, arguments must be added to justify chosen methodology.
  12. Line 350 – Amout of components in composition of film SCa5 is given in grams (g) in Figure 2, however in line 350 amount of CaCl2 for film SCa5 is given in % (0,02%). Please correct numbers, because 0,02 g from 5,02 g of total mass of film is not 0,02 %.
  13. What was the visual appearance of films? The addition of photos of obtained films would improve the understanding the manuscript and also complement the section of optical properties.
  14. Figure 4. Mark each separate figure by letters (e.g. a, b, c…).
  15. Figure 4. It is hard to understand the sence of absorbtion spectras in range 200-400 with such huge amount of peaks, furthermore somekind of numers are visible near the peaks, they must be explained. The visual information must be presented in better way.
  16. Figure 5. Letters and words are very poorly visible in the spectras. Consider combine some FTIR spectras, unify axes, if the aim is to compare them. Use the same color for substance in every spectra. Give reference to figure, when explain it in the text.
  17. Discussion part can be improved by giving references, comparing and telling more about what has been done and published in the field of mixing the same components for films by other authors. Are there publications about adding AA and CaCl2 in biopolymer films? In what amounts, concentration? What was the effect (in numbers) on properties of films etc. There are some references in Introduction without details, however it is not enough.

Round 2

Reviewer 1 Report

The authors made some revision. However the quality of Figures is still unsatisfactory. The title is very general and uninformative.

Reviewer 2 Report

The majority of questions and comments have been answered and the manuscript has been improved significantly.

It is hard to understand the changes in the presentation of UV and FTIR spectras. There are author’s comments and explanations, how pictures had been improved for better visual perceptibility, however because of number of old and new figures (showing document in track change mode), it is not easy to understand, which is new figures. However, I believe that authors have understand the significance of clear presentation of such data and final version will be easy to read and understand.

Discussion part has been improved by adding more references and explaining results comparing to other authors working in similar directions.

Round 3

Reviewer 1 Report

The title " Active Biodegradable 
3 Packaging Materials......" is still very gerneral and covers a vast area of reseach . Figure 6 is of very poor quality. Figure 6 consist of 12 figures nubmer 6a-f and 6g-l, which are not metionded in the text at all. 

Round 4

Reviewer 1 Report

The title is not content-specific, it is very general, if this can be the title of this submission, it can be the tiltes of a bunch of publications as well;

Figures are of poopr quality.